# Discovery of Potent and Selective Halogen-Substituted Imidazole-Thiosemicarbazides for Inhibition of *Toxoplasma gondii* Growth In Vitro via Structure-Based Design

**DOI:** 10.3390/molecules24081618

**Published:** 2019-04-24

**Authors:** Agata Paneth, Lidia Węglińska, Adrian Bekier, Edyta Stefaniszyn, Monika Wujec, Nazar Trotsko, Anna Hawrył, Miroslaw Hawrył, Katarzyna Dzitko

**Affiliations:** 1Department of Organic Chemistry, Medical University, Chodźki 4a, 20-093 Lublin, Poland; lidia.weglinska@umlub.pl (L.W.); edyta.stefaniszyn@umlub.pl (E.S.); monika.wujec@umlub.pl (M.W.); nazar.trotsko@umlub.pl (N.T.); 2Department of Immunoparasitology, Faculty of Biology and Environmental Protection, University of Lodz, Banacha 12/16, 90-237 Lodz, Poland; adrian.bekier@unilodz.eu; 3Department of Inorganic Chemistry, Medical University, Chodźki 4a, 20-093 Lublin, Poland; anna.hawryl@umlub.pl (A.H.); mirek.hawryl@umlub.pl (M.H.)

**Keywords:** thiosemicarbazides, anti-*Toxoplasma gondii* activity, cytotoxicity, selectivity ratio, structure-activity relationship (SAR) analysis

## Abstract

Employing a simple synthetic protocol, a series of highly effective halogen-substituted imidazole-thiosemicarbazides with anti-*Toxoplasma gondii* effects against the RH tachyzoites, much better than sulfadiazine, were obtained (IC_50s_ 10.30—113.45 µg/mL vs. ~2721.45 µg/mL). The most potent of them, 12, 13, and 15, blocked the in vitro proliferation of *T. gondii* more potently than trimethoprim (IC_50_ 12.13 µg/mL), as well. The results of lipophilicity studies collectively suggest that log*P* would be a rate-limiting factor for the anti-*Toxoplasma* activity of this class of compounds.

## 1. Introduction

*Toxoplasma gondii* is a common zoonotic infection of humans, and estimates prove that up to one third of the world’s population is chronically infected [1,2]. Although largely asymptomatic, chronic infection can be fatal or lead to serious problems in fetuses and immunocompromised patients [3,4,5,6,7,8,9]. Current administration of the therapeutics, like the combination of pyrimethamine and sulfadiazine, shows high rates of toxicity and side effects, including intolerance or allergic reaction to the sulfa component [10,11,12,13,14,15,16]. Other serious problems, such as the emergence of drug resistance and the incidence of relapses after discontinuation of therapy, in some cases are observed as well [17,18].

In this context, the thiosemicarbazide scaffold has emerged as a promising structure for the lead optimization process. In the search for new drug leads for toxoplasmosis, we are exploring the thiosemicarbazide scaffold as a promising lead structure for developing potent and selective anti-*Toxoplasma gondii* medicines. Preliminary screening of the imidazole-thiosemicarbazides has revealed several potent inhibitors of tachyzoite growth in vitro with much higher potency when compared to sulfadiazine [19]. Among them, the best anti-toxoplasma response was noted for those with electron-withdrawing nitro and chloro substitution at the N4 phenyl ring (Figure 1). Although new chemotypes were provided, low selectivity in the parasite inhibition over host cells, defined as the selectivity ratio of CC_50_ to IC_50_, was observed in most cases. Thus, there was a real need for new, more effective, less toxic, and thus more selective analogues. We were able successfully accomplish this goal by further exploiting the N4 phenyl position of the imidazole-thiosemicarbazide core with electron-withdrawing halogen substitution. In fact, based on our initial results, it is reasonable to suppose that the deactivation of the N4 phenyl ring, through the inductive withdrawing effect of halogen atoms, should result in compounds with potent activity against *T. gondii* tachyzoites growth. From the viewpoint of rational drug design, other factors, such as impact of halogenated compounds on membrane permeability, were of high importance. Indeed, for many years, the effective applied strategy for the hit-to-lead or lead-to-drug optimization process involved the insertion of halogens during the synthesis of final compounds [20,21,22]. This strategy is based on the observation that incorporation of the halogen atoms into a new bioactive chemical entity improves membrane permeability and oral absorption [23]. Further, halogenation enhances the blood-brain barrier permeability, which is a pre-requisite for drugs that need to reach the CNS, like anti-toxoplasma drugs and many others [24]. In this paper, employing the halogenation strategy, a series of highly effective halogenated-substituted imidazole-thiosemicarbazides, with much better anti-*Toxoplasma gondii* effects against the RH tachyzoites than sulfadiazine, were identified. The most potent of these imidazole-thiosemicarbazides blocked the in vitro proliferation of *T. gondii* more potently and selectively than pyrimethamine, as well. In further studies we show that the observed trend in their anti-*Toxoplasma gondii* activity depends significantly on the lipophilicity factor.

## 2. Results and Discussion

### 2.1. Molecular Design and Synthesis

As mentioned in the Introduction, in our previous study, a series of imidazole-thiosemicarbazides was tested to optimize compounds effective against tachyzoite proliferation [19]. We discovered that the variations at the N4 position of the thiosemicarbazide core led to the differentiation of the biological response. For example, compounds with the N4 aliphatic chain had poor activity (IC_50_ ≥ 125 µg/mL), while compounds with electron donating substitution at N4 aryl position were generally less potent than those with an electron withdrawing group. The best results for the inhibition of tachyzoites proliferation were obtained for the nitro derivatives **1** and **2** (Figure 1, left), and the difference with the control sulfadiazine was significant (IC_50_~2721.45 µg/mL). To better understand the structural and electronic determinants responsible for the observed trend in activity, a computational approach was subsequently performed; this approach led to the conclusion that the inductive withdrawing effect of substituents around the N4 phenyl ring, rather than its substitution pattern or geometry of molecules, are the key functionalities required for potent anti-*Toxoplasma gondii* activity. To test this assumption, a subsequent series of structural analogues of the nitro chemotypes **1** and **2** was designed and tested. Particularly, we investigated a panel of R groups in the context of the halogen substituents. A synthetic route for the preparation of the halogen-substituted series of imidazole-thiosemicarbazides **5**–**16** is presented in Scheme 1. The compounds were prepared under the routine protocol described elsewhere [25], in the one-step reaction of 4-methyl-imidazole-5-carbohydrazide with appropriate haloaryl isothiocyanate.

### 2.2. Cytotoxicity against L929 Cells

Since *T. gondii* is an obligate intracellular parasite that requires invasion of mammalian host cells to proliferate, compounds that inhibit parasite growth might be toxic to host cells as well. Therefore, as an initial indicator for host cell toxicity, we firstly evaluated compounds **5**–**16** for their ability to inhibit the growth of mouse fibroblast (L929) cells line according to the international standards: ISO 10993-5:2009(E). In Table 1, the results are presented as the percent of viable cells ± standard derivation in the concentrations range of **5**–**16** between 0 to 1000 µg/mL, together with the CC_50_ values.

To calculate the reduction of viability compared to the untreated blank the equation was used: viability (%) = 100% × sample OD_570_ (the mean value of the measured optical density of the treated cells)/blank OD_570_ (the mean value of the measured optical density of the untreated cells). CC_50_—represents the concentration of tested compounds that was required for a 50% cell proliferation inhibition in vitro. The effect of the tested compounds on the cell lines was measured using MTT assay according to the international standard: ISO 10993-5:2009(E). CC_50_ values were determined based on the plotted curves using GraphPad Prism program (version 8.0.1).

From the cytotoxicity assays against L929s, none of compounds **5**–**16** showed significant toxic effects on cells after 24 h of incubation. All of them were non-toxic at concentration less than ~195 µg/mL which makes them a good candidate for anti*-T. gondii* activity assay in vitro. From the point of view of rational drug design, two other important conclusions from these studies should also be mentioned. Firstly, the cytotoxic effect of **5**–**16** generally increases with halogen size from fluorine to iodine (F < Cl <Br < I). In other words, the observed cytotoxic effect increases when electronegativity of the halogen decreases. Secondly, like steric and electronic effects, the halogen distribution pattern has considerable effect on cytotoxicity. Clearly, except the fluoro isomers **5**–**7**, the general scale of toxicity is that *para*-isomer shows more toxicity than its *meta* counterpart, which in turn is more toxic than the isomer *ortho* (*para* > *meta* > *ortho*). Likewise, within a series of the fluoro isomers **5**–**7**, the *ortho*-isomer is the most toxic, followed by the *meta*-isomer, whereas the *para*-isomer is the least toxic.

### 2.3. In Vitro Activity and Cytotoxicity

Drug resistance in *Toxoplasma gondii* is a challenge not only for treatment failure and recurrent infections but also for the correct constructions of basic conclusions from the in vitro bioassays. The correct interpretation of the bioassay results, i.e., proof of effectiveness and proof of safety for the samples tested, depends, partly, on the drug susceptibility of a strain used in the experiments. Therefore, we started our biological studies with a prior determination of the resistance of the reference RH strain of *Toxoplasma* (ATCC^®^ PRA 310™) to sulfadiazine and trimethoprim; two routinely used positive controls used in the in vitro bioassays. As presented in Figure 2, this highly virulent RH strain is sulfadiazine resistant with an IC_50_ value higher than 2500 μg/mL. Trimethoprim was able to inhibit the tachyzoites production with a low concentration. However its cytotoxicity concentration (CC_50_ > 32.00 μg/mL) was unfavorably close to the inhibitory effect (IC_50_ = 12.13 μg/mL). Finally, only the combination of sulfadiazine and trimethoprim in the ratio 5:1 produced a non-toxic (CC_50_ = 2454.14 μg/mL) synergistic effect against *Toxoplasma gondii* growth in vitro, with an IC_50_ of 37.15 μg/mL.

Having determined the resistance of the reference RH strain to sulfadiazine and trimethoprim, we further investigated the efficacy of **5**–**16** in blocking *T. gondii* proliferation. Additionally, *T. gondii* proliferation was tested using a [^3^H]-uracil incorporation assay, which is specific for the labelling of the nucleic acids of the parasite. Sulfadiazine, trimethoprim and their combination in ratio 5:1, respectively, were used as positive controls, while DMSO (dimethyl sulfoxide) at a concentration of 0.1% was used as the negative control (data not shown). Results of the screen are summarized in Figure 3, where compounds are ranked by relative potency.

In these cellular assays, several prominent trends were observed. As shown in Figure 3 and Table 2, among the tested set of compounds **5**–**16**, all of them were more inhibitory to the growth of *T. gondii* than sulfadiazine. In addition, three of these, *meta*-bromo **12** (IC_50_ = 15.64 µg/mL), *para*-bromo **13** (IC_50_ = 14.57 µg/mL), and *meta*-iodo **15** (IC_50_ = 10.30 µg/mL) were of exceptionally high potency with growth IC_50_ values comparable to, or even better than, those of trimethoprim (IC_50_ = 12.13 µg/mL). Compounds with *ortho*-fluoro **5**, *meta*-fluoro **6**, *para*-fluoro **7**, and *para*-chloro **10** groups were the weakest inhibitors with IC_50_ at ~69 µg/mL or even higher.

Assessment of the results in terms of structural features leads to the following conclusions: (i) the inhibitory effect against *T. gondii* proliferation increases with halogen size from fluorine to iodine; compounds with bulky iodo and bromo groups were definitely more active than those with smaller chloro and fluoro substituents; (ii) the halogen distribution pattern has a considerable effect on the trends in the bioactivity. Clearly, except the bromo-substituted thiosemicarbazides **11**–**13**, the general scale of the inhibitory potency is that the *meta*-isomer shows more activity than its *ortho* counterpart which, in turn, is more effective than the isomer *para* (*meta* > *ortho* > *para*). In turn, within series **11**–**13** the *meta*-isomer is almost as effective as its *para* analogue (IC_50_~15 µg/mL), whereas the *ortho*-isomer is two times weaker in inhibitory action than its counterparts.

It is a well-established principle that compounds exhibiting large therapeutic windows between the inhibition of *T. gondii* proliferation and human cell growth are expected to be more effective and safer during in vivo treatment. In practice, the selectivity ratio, defined as the ratio of the 50% cytotoxic concentration (CC_50_) to the 50% antiparasitic concentration (IC_50_), is as widely used as the parameter to express a compound’s in vitro efficacy. Comparison of the test results in Table 2 shows our best inhibitor *meta*-I **15** to be more selective and more potent in inhibiting *T. gondii* cell growth in vitro than trimethoprim. This particular comparison favoring the *meta*-I **15** over trimethoprim in potency measure is not the only example favoring the halogen-substituted thiosemicarbazides listed in Table 2. For example, *meta*- and *para*-bromo derivatives **12** and **13** show higher selectivity ratios and only slightly lower inhibitory potency than control trimethoprim. Again, although the weakest compound in our study (**7**) is not particularly impressive in its inhibition of *T. gondii* proliferation, it still displays a more favorable selectivity ratio than those for trimethoprim.

### 2.4. Physicochemical Characterization of ***5***–***16***

Membrane permeability for a drug intended for an intracellular target is a key metric at the early stage of the anti-*Toxoplasma* drug development processes [26]. Indeed, a molecule with good in vitro activity but poor membrane permeability will have low or nonexistent efficacy in vivo. Therefore, a detailed understanding of the preferential partitioning of a drug candidate into the membrane is vitally important from a rational drug design standpoint. When lipophilicity and its relationship with passive drug transfer through physiological barriers has been extensively investigated, numerous significant correlations between log*P* and drug passive permeation have been established. In fact, absorption by passive diffusion permeation is generally considered optimal for compounds having a moderate log*P*, with a range between 1.5 to 2.7 for blood-brain barrier penetration and a range between 0 to 3 for optimal gastrointestinal absorption [27,28]. Compounds with a lower log*P* are more polar and have poorer lipid bilayer permeability while compounds with a higher log*P* are more lipophilic and thus have better membrane permeability. At the same time, however, serious liabilities for more lipophilic compounds can also be incurred, including poor water solubility, increased toxicity, and faster metabolic clearance [29].

In the light of all the facts mentioned above, for the purpose of the study, the lipophilicity of **5**–**16** was measured experimentally by the HPLC technique and its correlation with antiparasitic activity was probed. The log*P* parameters for **5**–**16** are shown in Table 2. The comparison of the log*P* and IC_50_ shows that the general trend in anti-*Toxoplasma gondii* activity of **5**–**16** is dependent upon lipophilicity and a significant incremental increase in the partition coefficient resulting from an increase in activity is observed, with the Spearman’s rank correlation coefficient 0.81 and *p* value 0.001 (Figure 4). In fact, except for **10**, the weakest inhibitors in our studies (**5**–**7**) are the least lipophilic, while the most lipophilic compound (**15**) proved to be the best inhibitor of *T. gondii* growth in vitro. Thus, the results collectively suggest that lipophilicity would be a rate-limiting factor for the anti-*Toxoplasma* activity of the halogen-substituted imidazole-thiosemicarbazides. From the point of view of rational drug design, other important conclusions from these studies should also be mentioned: (i) fluoro-substituted thiosemicarbazides (**5**–**7**) are the least lipophilic followed by the *ortho* isomers, **8**, **14**, **11**; (ii) lipophilicity of the *meta* and *para* isomers increases with halogen size from chlorine to iodine; (iii) *para* isomers tend to have lower lipophilicity than their *meta* counterparts (*para*-chloro < *meta*-chloro < *para*-bromo < *meta*-bromo < *para*-iodo < *meta*-iodo).

## 3. Materials and Methods

### 3.1. Chemistry

All commercial reactants and solvents were purchased from either Sigma-Aldrich (Saint Louis, MS, USA) or Alfa Aesar (Karlsruhe, Germany) with the highest purity and used without further purification. The melting points were determined on a Fischer-Johns block and are uncorrected. Analytical thin layer chromatography was performed with Merck (Darmstadt, Germany) 60F_254_ silica gel plates and visualized by UV irradiation (254 nm). Elemental analyses were determined by a AMZ-CHX elemental analyzer (PG, Gdańsk, Poland). Analyses indicated by the symbols of the elements were within ±0.4% of the theoretical values. The ^1^H-NMR were recorded on a Bruker Avance (300 MHz) spectrometer. For representative model compounds (**8**, **13**, and **16**) ^13^C-NMR spectra were also recorded on a Bruker Avance spectrometer. MS were recorded on a Bruker microTOF-Q II mass spectrometer (BioSpin, GmbH, Rheinstetten, Germany) using APCI method. The physicochemical characterization of compounds **5**–**7**, **9**, **10** were presented in [19,30,31].

### 3.2. Procedure for Synthesis of the Imidazole-Thiosemicarbazides ***5***–***16***

A solution of 4-methylimidazole-5-carbohydrazide (0.01 mol) and an equimolar amount of haloaryl isothiocyanate (0.01 mol) in 25 mL of anhydrous ethanol was heated under reflux for 10–30 min. After cooling, the solid formed was filtered off, dried, and crystallized from ethanol.

*4-(2-Chlorophenyl)-1-(4-methylimidazol-5-oyl)thiosemicarbazide* (**8**). Yield: 87%. M.p. 217–219 °C. ^1^H-NMR (DMSO-*d*_6_) δ (ppm): 2.43 (s, 3H, CH_3_); 7.19–7.25 (m, 1H, 1×CH_ar_); 7.30–7.35 (m, 1H, 1×CH_ar_); 7.45–7.53 (m, 1H, 1×CH_ar_); 7.75 (s, 1H, 1×CH_ar_); 8.80 (s, 1H, 1×CH_ar_); 9.31, 9.75, 9.92, 12.42 (4s, 4H; 4×NH). ^13^C-NMR (DMSO-*d*_6_) *δ* (ppm): 9.16, 125.59, 126.95, 127.24, 127.57, 127.83, 130.79, 130.89, 132.43, 135.05, 161.65, 180.01. MS (APCI) (*m*/*z*, %): = 170.0 (95), 141.1 (100), 127.1 (72), 109.0 (99). Anal. C_12_H_12_ClN_5_OS (C, H, N).

*4-(2-Bromophenyl)-1-(4-methylimidazol-5-oyl)thiosemicarbazide* (**11**). Yield: 94%. M.p. 205–207 °C. ^1^H-NMR (DMSO-*d*_6_) δ (ppm): 2.46 (s, 3H, CH_3_); 7.12–7.15 (m, 1H, 1×CH_ar_); 7.34–7.40 (m, 1H, 1×CH_ar_); 7.63 (s, 2H, 2×CH_ar_); 7.75–7.78 (d, 1H, 1×CH_ar_); 9.28, 9.75, 9.90, 12.42 (4s, 4H, 4×NH). MS (APCI) (*m*/*z*, %): = 213.9 (100), 174.0 (9), 141.1 (82), 127.1 (25), 109.0 (9). Anal. C_12_H_12_BrN_5_OS (C, H, N).

*4-(3-Bromophenyl)-1-(4-methylimidazol-5-oyl)thiosemicarbazide* (**12**). Yield: 82%. M.p. 204–206 °C. ^1^H-NMR (DMSO-*d*_6_) δ (ppm): 2.45 (s, 3H, CH_3_); 7.23–7.32 (m, 1H, 1×CH_ar_); 7.53–7.55 (m, 1H, 1×CH_ar_); 7.62 (s, 2H, 2×CH_ar_); 7.82 (s, 1H, 1×CH_ar_); 9.70; 9.79; 9.85; 12.40 (4s, 4H, 4×NH). MS (APCI) (*m*/*z*, %): = 213.9 (100), 174.0 (9), 141.1 (82), 127.1 (25), 109.0 (9). Anal. C_12_H_12_BrN_5_OS (C, H, N).

*4-(4-Bromophenyl)-1-(4-methylimidazol-5-oyl)thiosemicarbazide* (**13**). Yield: 78%. M.p. 228–230 °C. ^1^H-NMR (DMSO-*d*_6_) δ (ppm): 2.45 (s, 3H, CH_3_); 7.48 (s, 4H, 4×CH_ar_); 7.62 (s, 1H, 1×CH_ar_); 9.64; 9.78; 9.85; 12.40 (4s, 4H, 4×NH). ^13^C-NMR (DMSO-*d*_6_) *δ* (ppm): 19.13, 115.30, 125.79, 127.17, 129.41, 130.75, 132.29, 137.45, 161.46, 179.50. MS (APCI) (*m*/*z*, %): = 216.0 (96), 183.0 (64), 172.0 (45), 141.1 (100), 127.1 (56), 109.0 (63). Anal. C_12_H_12_BrN_5_OS (C, H, N).

*4-(2-Iodophenyl)-1-(4-methylimidazol-5-oyl)thiosemicarbazide* (**14**). Yield: 93%. M.p. 220–222 °C. ^1^H-NMR (DMSO-*d*_6_) δ (ppm): 2.46 (s, 3H, CH_3_); 6.95–7.01 (t, 1H, 1×CH_ar_); 7.35–7.41 (t, 1H, 1×CH_ar_); 7.60–7.62 (m, 2H, 2×CH_ar_); 7.83–7.86 (d, 1H, 1×CH_ar_); 9.24; 9.69; 9.85; 12.41 (4s, 4H, 4×NH). MS (APCI) (*m*/*z*, %): 262.0 (100), 220.0 (52), 183.0 (3), 141.1 (93), 127.1 (31), 109.0 (30), 93.1 (2). Anal. C_12_H_12_IN_5_OS (C, H, N).

*4-(3-Iodophenyl)-1-(4-methylimidazol-5-oyl)thiosemicarbazide* (**15**). Yield: 91%. M.p. 210–212 °C. ^1^H-NMR (DMSO-*d*_6_) δ (ppm): 2.45 (s, 3H, CH_3_); 7.08–7.13 (t, 1H, 1×CH_ar_); 7.45–7.48 (d, 1H, 1×CH_ar_); 7.55–7.58 (d, 1H, 1×CH_ar_); 7.62 (s, 1H, 1×CH_ar_); 7.93 (s, 1H, 1×CH_ar_); 9.65; 9.77; 12.40 (3s, 4H, 4×NH). MS (APCI) (*m*/*z*, %): 262.0 (100), 220.0 (52), 183.0 (3), 141.1 (93), 127.1 (31), 109.0 (30), 93.1 (2). Anal. C_12_H_12_IN_5_OS (C, H, N).

*4-(4-Iodophenyl)-1-(4-methylimidazol-5-oyl)thiosemicarbazide* (**16**). Yield: 95%. M.p. 225–227 °C. ^1^H-NMR (DMSO-*d*_6_) δ (ppm): 2.45 (s, 3H, CH_3_); 7.34–7.37 (d, 2H, 2×CH_ar_); 7.60–7.65 (d, 3H, 3×CH_ar_); 9.62; 9.77; 9.85; 12.40 (4s, 4H, 4×NH). ^13^C-NMR (DMSO-*d*_6_) *δ* (ppm): 9.17, 125.88, 127.13, 127.40, 130.78, 132.30, 135.27, 137.94, 160.31, 179.40. MS (APCI) (*m*/*z*, %): 262.0 (100), 220.0 (42), 183.0 (17), 141.1 (92), 127.1 (32), 109.0 (30), 93.1 (2). Anal. C_12_H_12_IN_5_OS (C, H, N).

### 3.3. Assay In Vitro for Anti-T. gondii Activity

Preparation of compounds and drugs: the dilution of all compounds and drugs was freshly prepared before the cells or *T. gondii* were exposed. Compounds **5**–**16** were dissolved in dimethyl sulfoxide (DMSO, Sigma, Saint Louis, MS, USA) to 50 mg/mL and the final concentration of solvent was not higher than 2.00%. Sulfadiazine [4-amino-*N*-(2-pyrimidinyl)benzenesulfonamide] (S8626, Sigma) was dissolved in 1M sodium hydroxide (NaOH, Sigma) to 100 mg/mL and the final concertation of NaOH was not higher than 0.25% in the tested concentrations. Trimethoprim [2,4-diamino-5-(3,4,5-trimethoxybenzyl)pyrimidine] (92131, Sigma) was dissolved in DMSO to 50 mg/mL, and the final concertation of DMSO in trimethoprim dilutions was not higher than 0.32%. Sulfadiazine and trimethoprim with a ratio of 5:1 (Sul-Tridin 24%, 200 mg/mL + 40 mg/mL, ScanVet, Warsaw, Poland) ready solution was diluted in appropriate medium.

Cell lines and parasite culture: the Hs27 (human foreskin fibroblast) (ATCC^®^ CRL-1634™, Manassas, VA, USA) were maintained according to the ATCC product sheet. Shortly, cells were cultured in DMEM (DMEM/10%FBS/P/S) (Dulbecco’s Modified Eagle’s Medium—ATCC^®^ 30-2002™) culture medium supplemented with 10% FBS (Fetal Bovine Serum, ATCC^®^ 30-2020™), 100 U/mL penicillin, and 100 μg/mL streptomycin (Penicillin-Streptomycin Solution (ATCC^®^ 30-2300™). The cell line, when it achieved a confluent monolayer, was trypsinized (using a Trypsin-EDTA Solution, 1X ATCC^®^ 30-2101™) and seeded at a density of 1 × 10^6^ per T25 cell culture flask (Falcon) and incubated for 48–72 h in a 37 °C and with 5 CO_2_. The RH strain of *Toxoplasma gondii* (ATCC^®^ PRA 310™) is highly virulent and belongs to the first haplogroup. Tachyzoites were maintained according to the ATCC product sheet, in parasite culture medium (DMEM/3%HIFBS/P/S), which contains DMEM medium with 3% HIFBS (Heat-Inactivated (1 h in 56 °C) Fetal Bovine Serum, ATCC^®^ 30-2020™). Infected tissue culture cells were incubated in a 37 °C and 5% CO_2_.

Influence of **5**–**16** and drugs on *T. gondii* proliferation: 1 × 10^4^ per well of the Hs27 cells were seeded on 96-well plates in DMEM/10%FBS/P/S. After 72 h of incubation, the medium was removed and then tachyzoites of the RH strain were added to the cell monolayers, MOI 10, in DMEM/3%HIFBS/P/S. One hour later, different compounds or drugs (sulfadiazine, trimethoprim and sulfadiazine with trimethoprim) dilutions (100 µL/well) in the parasite culture medium were added to the cell monolayers with *T. gondii*. After a subsequent 24 h of incubation, 1 µCi/well [5,6-^3^H] uracil (Moravek Biochemicals Inc., Brea, CA, USA) was added to each microculture for a further 72 h. The amount of the isotope incorporated into the parasite nucleic acid pool, corresponding to the parasite growth, was measured by liquid scintillation counting using 1450 Microbeta Plus Liquid Scintillation Counter (Wallac Oy, Turku, Finland). The results were expressed as counts per minute (CPM) and transformed to the percentage of viability compared to untreated cells. All experiments were performed in triplicate.

Since direct comparison of IC_50_ can be difficult when assays are performed with different strains at different times, two compounds known from our previous studies (**6** and **9**) [14] were also included in the bioassay performed for compounds **5**, **7**, **8**, **9**–**16**.

### 3.4. Cytotoxic Assay

Cell culture: the L929 mouse fibroblast (ATTC^®^ CCL-1™) was routinely cultured in IMDM (Iscove’s Modified Dulbecco Medium, Biowest, Nuaille, France), supplemented with 10% HIFBS, 100 U/mL penicillin and 100 μg/mL streptomycin. Cells were trypsinized twice a week and seeded at a density of 1 × 10^6^ per T25 cell culture flask and incubated for 24–48 h in a 37 °C and 10% CO_2_ to achieve a confluent monolayer.

Cell viability assay: Assays were performed according to international standards: ISO 10993-5:2009(E), Biological evaluation of medical devices, Part 5: Tests for in vitro cytotoxicity. Cell viability were evaluated using the tetrazolium salt [3-(4,5-dimethylthiazol-2-yl)-2,5-diphenyltetrazolium bromide] (Sigma) and mouse fibroblasts L929 cells. For this assay we used only the culture medium, RPMI 1640, without phenol red (Biowest) supplemented with 10% HIFBS, 2 mM L-glutamine (Sigma), 100 U/mL penicillin, 100 μg/mL streptomycin. Shortly after, a 1 × 10^4^/well of L929 cells were placed into 96-well plates and incubated in 37 °C and 10% CO_2_. Afterwards, an old culture medium was replaced by 100 μL of compounds and drugs dilutions in culture medium and the cells were treated for 24 h. Moreover, cells were treated with a 4.0–0.03% concentration of DMSO and 1M NaOH as a compound and drugs solvent (data not shown). Then, 50 μL of 1 mg/mL of MTT solution in RPMI 1640 without phenol red was added to each well and incubated 2h (37 °C, 10% CO_2_). Next, cell culture medium was aspirated cautiously and 150 μL of DMSO was added to each well, the plates were gently mixed. Then 25 μL 0.1M glycine buffer (pH = 10.5) (Sigma) was added. The optical density at 570 nm on the ELISA reader (Multiskan EX, Labsystems, Vienna, VA, USA), was read. The results were expressed as a percentage of viability compared to untreated cells. All experiments were performed in triplicate. For comparison reasons, two compounds known from our previous studies (**6** and **9**) [14] were included in the bioassay performed for compounds **5**, **7**, **8**, **9**–**16**.

### 3.5. Graphs and Statistical Analyses

Statistical analyses and graphs were performed using GraphPad Prism version 8.0.1 for macOS (GraphPad Software, San Diego, CA, USA). For compounds with CC_50_ or IC_50_ values greater than the highest concentration tested, values were calculated based on extrapolation of the curves using the GraphPad Prism program (version 8.0.1). Additionally, to establish a relationship between cytotoxicity and antiparasitic activity, the selectivity ratio (SR) values were calculated as the ratio of the 50% cytotoxic concentration (CC_50_) to the 50% antiparasitic concentration (IC_50_). The higher the SR value the more selective the compounds were towards inhibiting *T. gondii* proliferation vs. cytotoxicity.

### 3.6. Lipophilicity Studies

HPLC experiments were performed using a chromatograph equipped with an Elite LaChrom L-2130 gradient pump, L-2455 DAD detector, L-2300 column oven, and L-2200 autosampler (Hitachi-Merck, Darmstadt, Germany). The tested samples were applied into the Phenomenex Synergi 4 µm Polar-RP chromatographic column (Phenomenex, Torrance, CA, USA) (150 mm length, 4.6 mm i.d.). Mobile phases were degassed using a built-in membrane degasser.

Chromatographic analysis: 5 µL of each sample (0.1% methanolic solutions) was applied into the chromatographic column and chromatograms were developed at a flow rate of 1.0 mL min^−1^ in the isocratic mode using various concentrations of methanol in the binary polar mobile phase: methanol:water (40:60, 45:55, 50:50, 55:45, 60:40, 65:35, and 70:30 (*v*/*v*)) Chromatograms were detected at 254 nm. All experiments were repeated in triplicate, and the final results were their arithmetic mean. Dead time was measured for uracil (Calbiochem—Merck, Darmstadt, Germany). All the experiments were performed at ambient temperature.

The chromatographic lipophilicity parameter (log*k_w_*) for **5**–**16** was obtained by the extrapolation of the retention parameter log*k* to pure water, according to the equation:

log*k* = log*k_w_* − *S* × *φ*(1)
where log*k_w_* is the value of the retention factor of a substance in pure water, *S* is the slope of the regression curve, and *φ* is the concentration (*v*/*v*) of methanol in the mobile phase.

Subsequently, log*P*_HPLC_ parameters for **5**–**16** were calculated according to OECD guidelines [32], based on the equation obtained for seven standards with known lipohilicity parameters (log*P_o/w_*):

log*P_o/w_* = 1.0239(± 0.03)·log*k_w_* − 0.1541(± 0.08); n = 7; r = 0.9976; s_e_ = 0.08; F = 1047.8(2)

The log*P_o/w_* values for seven standards used in experiments, i.e., aniline (0.9), 2-hydroxyquinoline (1.26), benzene (2.1), toluene (2.7), bromobenzene (3.0), naphtalene (3.6), and ethylbenzene (3.2) were analyzed in the same chromatographic conditions as compounds **5**–**16**.

## 4. Conclusions

We evaluated a series of halogen-substituted imidazole-thiosemicarbazides for their ability to inhibit in vitro growth of the apicomplexan parasite *T. gondii*. All compounds blocked in vitro proliferation of *T. gondii* more potently than sulfadiazine, with the estimated IC_50_ values in the range of 10.30–113.45 µg/mL vs. ~2721.45 µg/mL for sulfadiazine. The most potent of them with *meta*-bromo **12** (IC_50_ 15.64 µg/mL, SR 15.2), *para*-bromo **13** (IC_50_ 14.57 µg/mL, SR 15.35), and *meta*-iodo **15** (IC_50_ 10.30 µg/mL, SR 19.16) substitutions were at least comparable active and at the same time more selective than trimethoprim (IC_50_ 12.13 µg/mL, SR > 2.64) as well. All derivatives with fluoro substitution were found as the weakest inhibitors, suggesting that such type of substitution is not crucial for anti-toxoplasma activity of imidazole-thiosemicarbazides. Finally, the correlation of log*P* with antiparasitic activity was probed. The results of these studies collectively suggest that lipophilicity would be a rate-limiting factor for the anti-*Toxoplasma* activity of the halogen-substituted imidazole-thiosemicarbazides. Ongoing work is focused on the in vivo studies of the most potent halogen-substituted imidazole-thiosemicarbazides in parasitic challenge models in mice.

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
