# Peer review of "Discovery of Potent and Selective Halogen-Substituted Imidazole-Thiosemicarbazides for Inhibition of Toxoplasma gondii Growth In Vitro via Structure-Based Design"

_molecules, 2019, doi:10.3390/molecules24081618_

Round 1

Reviewer 1 Report

The authors propose a new conditional drug for the treatment of toxoplasmosis, but many new experiments are needed to demonstrate the proposed drug's efficacy. When using the RH strain at the maximum we evaluate is the acute phase of infection that is not so long in the case of this disease. The major challenge in toxoplasmosis is to find a drug that attacks tissue cysts and promotes the combat against parasites in the chronic phase of the disease. Many studies have yet to be done to actually make a proposal for a substitute drug to those already in existence.

Author Response

We agree with the Reviewer that many new experiments are needed to demonstrate the proposed drug's efficacy. The results obtained in this research are the basis for the currently started experiments in our lab including in vivo tests on a mouse model with another T. gondii strain (Me49 and DX) responsible for cyst formation. If they will be found to be effective in in vivo studies, such compounds could represent leads in the development of novel drugs against toxoplasmosis. From the other hand, due to slight differences between tachyzoite and bradyzoite, such as alterations to metabolism, morphological changes and stage-specific antigen expression, we can speculate that our compounds have also activity against bradyzoites.

Reviewer 2 Report

1.       The manuscript should be better foccused and much shorter. Introduction is too long for such type of experimental paper, Results and discussion part also could be shortened.

2.       Novelty of the chapter 2.3 is not high enough. Results presented in this chapter could be merged with the results presented in chapter 2.4.

3.       Description of types of the molecular transport in chapter 2.5 is too long and there is no clear link to the results presented.

4.       Last two columns (TPSA and %ABS) In Table 3 of do not contain valuable information.  

5.       Conclusions are not informative, differences in the anti-toxoplasmal properties of the synthesized compounds are not properly discussed.

6.       Only last two sentences of the abstract contain information on the results of manuscript, but they also are almost „empty“. The MS contains rather interesting results and it is not enough to say that the synthesized halogen-substituted imidazole-thiosemicarbazides are better anti-toxoplasmal compounds then sulfadiazine and  pyrimethamine. 12 interesting compounds were synthesized and conclusions on their structure-functions relationship should be better presented

7.     Structures of the synthesized compounds presented in Table 2 should be of much higher quality. 

Author Response

Reviewer 2:

The manuscript should be better focused and much shorter. Introduction is too long for such type of experimental paper, Results and discussion part also could be shortened.

Introduction and Results and Discussion have been shortened as suggested. Particularly, the results presented in chapter 2.3 has been merged with those in chapter 2.4. Moreover, the chapter 2.4 (originally chapter 2.5) has been modified; firstly, since the Table 3 was not especially informative, it was removed from the manuscript, secondly, all discussion on TPSA and %ABS that did not contain valuable information together with description of types of the molecular transport were also removed from the manuscript.

Novelty of the chapter 2.3 is not high enough. Results presented in this chapter could be merged with the results presented in chapter 2.4.

The results presented in chapter 2.3 has been merged with those in chapter 2.4.

Description of types of the molecular transport in chapter 2.5 is too long and there is no clear link to the results presented.

Description of types of the molecular transport in chapter 2.5 (now in chapter 2.4) was removed from the manuscript.

Last two columns (TPSA and %ABS) In Table 3 of do not contain valuable information.

Table 3 was removed from the manuscript. All discussion on TPSA and %ABS that did not contain valuable information was also removed from the manuscript.

Conclusions are not informative, differences in the anti-toxoplasmal properties of the synthesized compounds are not properly discussed.

Conclusions has been modified.

Only last two sentences of the abstract contain information on the results of manuscript, but they also are almost „empty“. The MS contains rather interesting results and it is not enough to say that the synthesized halogen-substituted imidazole-thiosemicarbazides are better anti-toxoplasmal compounds then sulfadiazine and pyrimethamine. 12 interesting compounds were synthesized and conclusions on their structure-functions relationship should be better presented

Abstract has been modified.

Structures of the synthesized compounds presented in Table 2 should be of much higher quality.

Structures have been improved.